# Upregulation of neurovascular communication through filamin abrogation promotes ectopic periventricular neurogenesis

Shauna L Houlihan[1,2,3†], Alison A Lanctot[1,2,3†], Yan Guo[1,2], Yuanyi Feng[1,2*]

[1]Department of Neurology, Northwestern University Feinberg School of Medicine, Chicago, United States; [2]Center for Genetic Medicine, Northwestern University Feinberg School of Medicine, Chicago, United States; [3]Driskill Graduate Program, Northwestern University Feinberg School of Medicine, Chicago, United States

**Abstract** Neuronal fate-restricted intermediate progenitors (IPs) are derived from the multipotent radial glia (RGs) and serve as the direct precursors for cerebral cortical neurons, but factors that control their neurogenic plasticity remain elusive. Here we report that IPs' neuron production is enhanced by abrogating filamin function, leading to the generation of periventricular neurons independent of normal neocortical neurogenesis and neuronal migration. Loss of Flna in neural progenitor cells (NPCs) led RGs to undergo changes resembling epithelial-mesenchymal transition (EMT) along with exuberant angiogenesis that together changed the microenvironment and increased neurogenesis of IPs. We show that by collaborating with β-arrestin, Flna maintains the homeostatic signaling between the vasculature and NPCs, and loss of this function results in escalated Vegfa and Igf2 signaling, which exacerbates both EMT and angiogenesis to further potentiate IPs' neurogenesis. These results suggest that the neurogenic potential of IPs may be boosted in vivo by manipulating Flna-mediated neurovascular communication.

*For correspondence: yuanyi-feng@northwestern.edu

†These authors contributed equally to this work

Competing interests: The authors declare that no competing interests exist.

## Introduction

Neurons in the cerebral cortex are generated from a hierarchically organized pool of neural progenitor cells (NPCs) distinctively characterized by the essential presence of neuronal fate-restricted intermediate progenitors (IPs) (*Laguesse et al., 2015*; *Lui et al., 2011*). Derived from the apical-basally polarized epithelial-like multipotent neuroepithelial (NE) or radial glial (RG) progenitors of the ventricular zone (VZ), IPs constitute a heterogeneous cell population, undergo substantial evolutionary expansion, and contribute significantly to both increased number and function of cortical neurons (*Hansen et al., 2010*; *Kriegstein et al., 2006*). However, IPs are believed to have limited proliferation potential and can only divide once or twice before generating excitatory neurons that migrate radially to the cortical plate (CP) (*Noctor et al., 2004*; *Taverna et al., 2014*). Therefore, enhancing the transit amplification and neurogenic potential of IPs would exclusively promote neuron production, but the mechanisms that control the plasticity of IPs remain unidentified.

In contrast to the multipotent NPCs of the VZ, IPs reside distantly from the lateral ventricles and are exposed to a distinctive microenvironment in the subventricular zone (SVZ), where they contact intimately with both the basolateral surface of RGs and blood vessels (*Javaherian and Kriegstein, 2009*; *Stubbs et al., 2009*). In the developing cerebral cortex, vascular circulation provides key factors to support NPC growth and differentiation. Conversely, VEGF-A produced by NPCs is indispensable for angiogenesis (*Ogunshola et al., 2002*; *Wittko-Schneider et al., 2014*). The extensive

communication between the vasculature and NPCs thereby permits the orchestration of angiogenesis with neurogenesis (*Vasudevan et al., 2008*). However, the molecular control responsible for NPC-blood vessel congruency is yet to be defined; it is also unclear whether the neurogenic potential of IPs may be boosted by angiogenic manipulation.

The X-linked *FLNA* is essential for the development of both the cerebral cortex and cardiovascular system. Females who lose a copy of *FLNA* present Periventricular Nodular Heterotopia (PH or PVH), a disorder that manifests as grossly normal-appearing cerebral cortex but ectopically placed nodular gray matter along the lateral ventricles (*Eksioglu et al., 1996*; *Fox et al., 1998*; *Parrini et al., 2006*; *Sheen et al., 2001*), while males carrying germline *FLNA* mutations often die prenatally of severe hemorrhage and cardiovascular defects (*Bernstein et al., 2011*; *Eksioglu et al., 1996*; *Reinstein et al., 2013*). As *FLNA* encodes the 280 kDa filamin A (FLNA) that together with FLNB and FLNC constitutes a family of large actin binding proteins, PH associated with *FLNA* loss-of-function mutations has been thought to be caused by an X-chromosome inactivation-mediated mosaic neuronal migration arrest, which assumed cortical neurons that inherited the mutant *FLNA* allele lacked actin promoted cell motility. However, radiological and pathological findings in both females and a rare case of a male infant with inherited *FLNA* mutations showed unremarkable neocortical size and structural aberration, (*Ferland and Guerrini, 2009*; *Guerrini et al., 2004*; *Parrini et al., 2011*; *Poussaint et al., 2000*; *Reinstein et al., 2012*), implying more complex mechanisms than the actin based mechanical failure. Interestingly, analysis of postmortem brains with *FLNA* mutations reported microvascular anomalies in addition to PH (*Ferland and Guerrini, 2009*; *Kakita et al., 2002*), suggesting the tied requirement of FLNA for cerebral cortical neural and vascular development. Similar to *FLNA* loss-of-function in males, mice with an engineered null mutation of *Flna* die embryonically of widespread hemorrhage that was accompanied by aberrant cardiovascular morphogenesis and cerebral cortical angiogenesis but not by neuronal migration defects (*Feng et al., 2006*).

To reveal the mechanism of *FLNA* in cortical development, we conditionally abrogated Flna in NPCs and developed a mouse model that highly resembles PH associated with *FLNA* loss-of-function in humans. We demonstrate in this study that *FLNA*-associated PH primarily results from the increased neurogenesis of IPs ectopically at no expense to the number and laminar organization of neocortical neurons. Our data show that Flna acts synergistically with β-arrestin and serves as a cytoplasmic break to regulate the homeostatic communication between NPCs and the cerebral vasculature, and loss of this function results in failed dampening of growth signals between NPCs and blood vessels, leading to excessive angiogenesis and sustained epithelial-mesenchymal transition (EMT)-like changes in RGs, which altogether alter IPs' microenvironment and increase their proliferation and neurogenesis. These findings reveal the dependence of FLNA in coordinated cerebral neurogenesis and vascularization and demonstrate the vascular impact on IPs' neurogenic plasticity.

## Results

### Loss of filamin results in PH without affecting neocortical neurons

We assessed the role of FLNA in cortical development by conditionally abrogating filamin A (Flna) in NPCs using either *Emx1*-Cre or *Nes*-Cre lines (*Gorski et al., 2002*; *Petersen et al., 2004*). As *Flna* conditional mutants (*Flna*$^{flox/y:Cre+}$, referred to as Flna$^{cKO}$) only showed mild phenotype in neuroependyma defects (data not shown),we considered the co-expression and possible redundant function of Flna and filamin B (Flnb) in NPCs (*Sheen et al., 2002*), generated compound mutants of Flna$^{cKO}$ and *Flnb*, and found the abnormal presence of nodular neurons near lateral ventricles in the brain of these mutants (*Figure 1A,B*; *Figure 1—figure supplement 1A–F*). All Flna$^{cKO}$; *Flnb*$^{-/-}$ mice and approximately 20% of Flna$^{cKO}$; *Flnb*$^{+/-}$ mice (referred to collectively as Fln$^{cKO-NPC}$ hereafter unless noted specifically) showed this phenotype. The periventricular neuronal nodules (referred to as PH hereafter) were variable in size, number, and distribution but presented exclusively in both cerebral hemispheres with mixed neurons and glia (*Figure 1B*). We detected PH at birth (*Figure 1—figure supplement 1G*) and found that the occurrence of PH was spatially associated with disruption of the neuroependymal lining (*Figure 1C*). Most neurons in the PH expressed *Cux1*, which defines late-born excitatory neurons in upper cortical layers (*Figure 1D*, *Figure 1—figure supplement 1H*), whereas very few PH neurons expressed *Foxp2*, which marks early-born neurons in the deep cortical

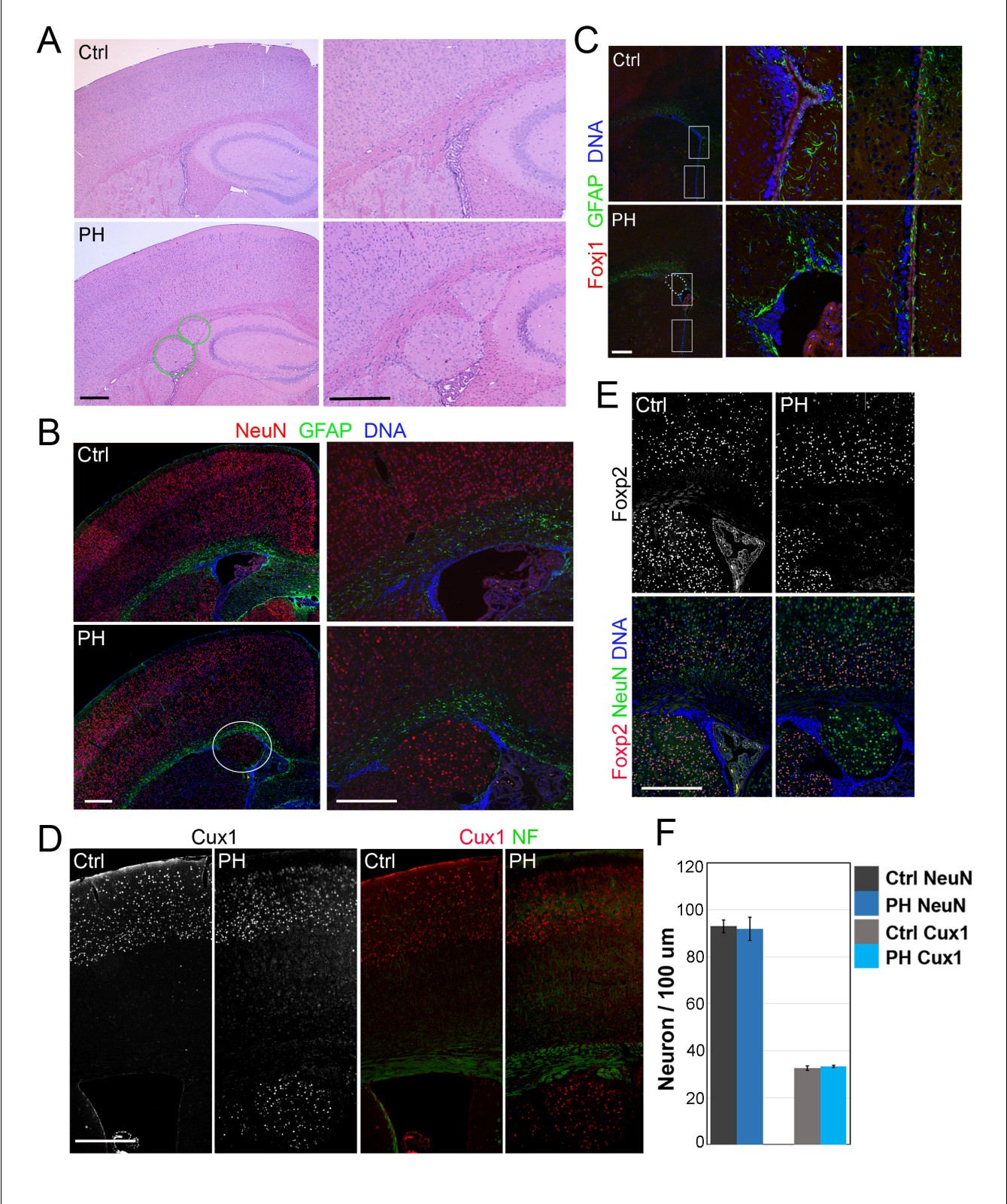

**Figure 1.** Loss of filamin resulted in Periventricular heterotopia without affecting neocortical neurons. (**A**) H&E stained coronal brain sections from a control (*Flna^flox/y Emx1Cre-*;*Flnb^+/-*·Ctrl) and a *Flna^flox/y Emx1Cre+* ;*Flnb^-/-* (PH) mouse at 2 month (P60). Periventricular heterotopia are indicated by circles; and shown by higher magnification views. (**B**) NeuN (red) and GFAP (green) double immunostained brain sections from a control and a *Flna^flox/y Emx1Cre+* ; *Flnb^-/-* (PH) mouse at weaning age (P23). Periventricular heterotopia are indicated by circles and are shown in higher magnification images. (**C**)

*Figure 1 continued on next page*

*Figure 1 continued*

FoxJ1 and GFAP double immunostained brain sections from a control and a *Flna*$^{flox/y\ Emx1Cre+}$ *;Flnb*$^{+/-}$(PH) mouse at weaning age. Higher magnifications of boxed areas shown on the right indicate that the loss of ependymal cells was specifically associated with the presence of periventricular heterotopia. (D) Cux1 and Neurofilament (NF) double immunostained cortical sections from a control and a *Flna*$^{flox/y\ Emx1Cre+}$ *;Flnb*$^{+/-}$ (PH) mouse at weaning age. (E) Foxp2 and NeuN double immunostained cortical sections from a control and a *Flna*$^{flox/y\ Emx1Cre+}$ *;Flnb*$^{+/-}$(PH) mouse at weaning age. (F) Quantitative analysis of neurons in the neocortex of post weaning age control and Fln$^{cKO-NPC}$ brains. All NeuN+ or Cux1+ cells between pial surface and white matter tracts were scored and normalized to the length of ventricular surface in the counted area. Data were obtained from brain sections of four pairs of control and Fln$^{cKO-NPC}$ littermates and presented as Mean ± SD. No significant difference was found by Student's t-test. Note the normal neocortical thickness and lamina organization in all brains with periventricular heterotopia. Bars: 500 µm.

The following figure supplement is available for figure 1:

**Figure supplement 1.** Undetectable neocortical aberration in PH-containing Fln$^{cKO-NPC}$ brains.

layer (*Figure 1E*; *Figure 1—figure supplement 1I,J*). Despite the presence of a large number of periventricular neurons, the neocortex of Fln$^{cKO-NPC}$ mice was well laminated and showed no change in either thickness or the number of cortical neurons relative to that of *Flna*$^{flox/y:Cre-}$; *Flnb*$^{+/-}$ littermate controls (referred to as Ctrl hereafter unless noted specifically) (*Figure 1F*; *Figure 1—figure supplement 1H–J*). These findings are all consistent with radiological and pathological reports of PH patients with *FLNA* mutations, and they suggest that PH caused by filamin loss is a unique condition in which additional neurons were generated at no expense of neocortical structure.

## A surplus in cortical neuron generation caused by filamin deficiency

To validate the association of PH with ectopically increased neurogenesis, we determined the developmental origin of PH and assessed cortical neuronal migration in Fln$^{cKO-NPC}$ brains though a BrdU birthdate study. Our analyses of both number and distribution of neurons born at E12.5, E14.5, or E16.5 did not show a significant difference between the neocortices of PH-containing Fln$^{cKO-NPC}$ brains and those of the control brains (*Figure 2A–D*; *Figure 2—figure supplement 1A,B*). This indicated that PH was not an outcome of failed neocortical neuronal migration but rather represented a net gain of neurons at the periventricular space.

Because excitatory neurons that constitute PH are known to be generated in situ from the radially organized cortical columns as opposed to coming in tangentially, we assessed the gained neurogenesis in PH-associated cortical columns by comparing the total number of neurons produced in these columns with cortical columns in adjacent PH-lacking regions as well as in anatomically matched control littermates. We found significantly more neurons were born at both E14.5 and E16.5 in PH-associated columns (total number of BrdU+ neurons in the entire depth of cortical wall normalized to the length of the ventricles), whereas BrdU+ cells in cortical columns lacking PH of the same mutant brain were not different from those in controls (*Figure 2E*). To rule out the possible mosaicism of Flna$^{cKO}$, we confirmed the effective conditional abrogation of Flna by *Emx1*$^{Cre}$ and showed that Flna protein was absent in all NPC-derived cortical cells of the Fln$^{cKO-NPC}$ embryos by E12.5, though it was highly detectable in cortical vascular cells of these embryos (*Figure 3—figure supplement 1A*). Together, these data confirmed the increased regional neurogenesis concomitant with the normal generation and migration of neocortical neurons in PH-containing Fln$^{cKO-NPC}$ brains; they also suggested that neurons in PH were produced as a surplus by an NPC population distinctive from those that gave rise to neurons destined for the neocortex.

## Increased IP neurogenesis at the periventricular space

To identify NPCs responsible for PH formation, we examined embryonic development of Fln$^{cKO-NPC}$ cortices. While none of the compound mutants showed a discernible phenotype in NPC number and proliferation rate or structural organization of the VZ, SVZ, and CP at E12.5 (*Figure 3—figure supplement 1B–D*), we started to observe disrupted adherence junctions (AJ) in a small number of RGs in Flna$^{cKO-NPC}$; *Flnb*$^{-/-}$ embryos at E13.5 (*Figure 3A*, *Figure 3—figure supplement 2A*). The AJ defect in Flna$^{cKO-NPC}$; *Flnb*$^{-/-}$ embryos was an isolated event but very frequently coincided with the mislocalization of neighboring IPs (marked by Tbr2) to the apical side of the VZ (*Figure 3A*, arrow). As development proceeded to mid-late stages of cortical neurogenesis, these phenotypes became

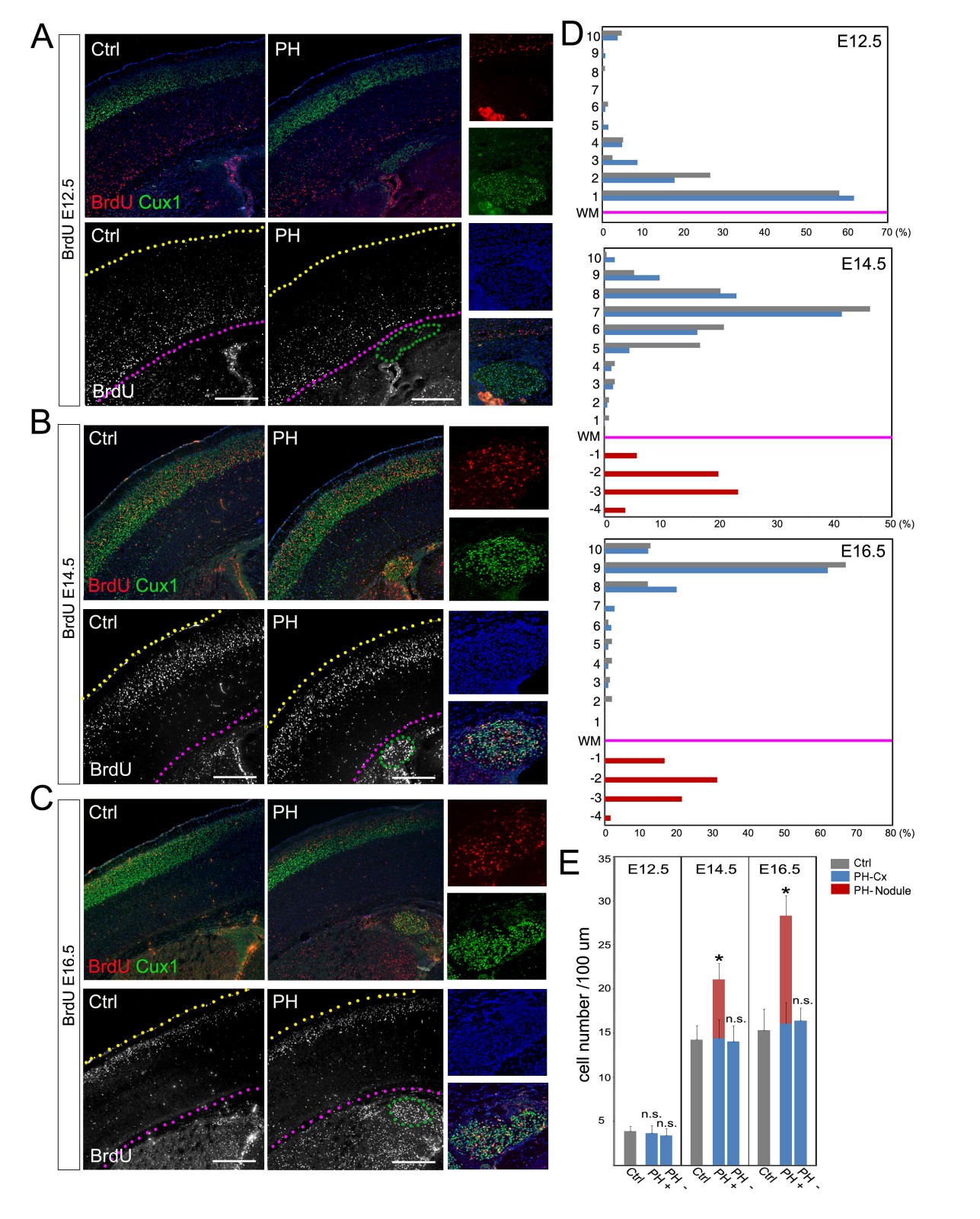

**Figure 2.** BrdU birth-dating study of cortical neuronal migration and neurogenesis. (**A–C**) A single pulse of BrdU was given at E12.5, E14.5 or E16.5 to label cells undergoing terminal mitosis. Brain sections from at least three sets of BrdU labeled littermates were analyzed at P4 by double immunohistological staining using antibodies to BrdU (red) and Cux1 (green). PH nodules were identified by Cux1+ immunoreactivity beneath the white matter tract. The positions of pial surface and white matter tract are marked by yellow and pink dotted lines, respectively. High magnification views
*Figure 2 continued on next page*

*Figure 2 continued*

show the lack of BrdU+ neurons in PH born at E12.5 but abundant BrdU+ neurons in PH born at E14.5 and E16.5. (D) Quantitative analysis of the distribution of BrdU labeled neurons in the brain of Fln$^{cKO-NPC}$ (PH) and their control (Ctrl) littermates. The neocortex of each brain section was divided into 10 equal layers from the pial surface to the white matter tract; the fraction of (%) of BrdU+ neurons in each layer was plotted to represent the relative distance of neuronal migration from the ventricular zone to the cortical plate. Each experiment was repeated with at least 3 litters; the result from a representative litter is presented. No significant difference in the distribution of cortical neurons was found between PH and Ctrl groups by Chi-square test. The number and distribution of BrdU+ neurons in PH (fractioned relatively to those in the neocortex) are also presented. (E) Quantitative analysis of neurons born at E12.5, E14.5 or E16.5 in filamin mutants (PH) and control brains. Cortical regions with and without the occurrence of PH in Fln$^{cKO-NPC}$ brains were scored separately; the total number of BrdU+ neurons in the entire cortical wall along 100 micron of white matter tract was scored. Data were acquired from at least three independent litters and presented as Mean + SD; *p<0.001 by ANOVA. Nuclei DNA was stained with Hoechst 33,342 and shown in blue in all fluorescence images. Bars: 500 µm.

The following figure supplement is available for figure 2:

**Figure supplement 1.** BrdU birth-dating studies of cortical neurogenesis and neuronal migration.

more pronounced and were seen in both Flna$^{cKO-NPC}$;*Flnb*$^{-/-}$ and Flna$^{cKO-NPC}$; *Flnb*$^{+/-}$ mutants (also referred to as PH hereafter). However, the apical mislocalization of IPs was exclusively associated with the disassembly of AJs and remained sporadic without affecting the nearby cortical tissue (*Figure 3B–D*; *Figure 3—figure supplement 2A–C*). The apicalized IPs incorporated BrdU robustly, divided near the ventricular surface, and produced neurons ectopically (*Figure 3E–H*; *Figure 3—figure supplement 2A–D,H*). In contrast, Pax6 positively labeled RGs in the affected regions became scarce and showed decreased mitotic activities (*Figure 3—figure supplement 2E–G*). Despite the severe disorganization of VZ and SVZ, the bulk of neurons destined for the neocortex and the structure of cortical plate were not interfered with (*Figure 3G–I*, *Figure 3—figure supplement 2D,H*). This finding was consistent with the lack of neocortical defect in PH brains and suggested that mislocalization of IPs to the periventricular space rendered them a more stimulative microenvironment, which expanded the IP pool and increased their neuronal production without affecting those IPs that were already fate-committed to generate neocortical neurons.

To test that apicalized IPs acquired higher potential in transient amplification and neurogenesis, we compared Tbr2+ IPs in cortical regions with apicalized IPs to adjacent regions where IPs reside normally as well as to spatially matched cortical regions of littermate controls. Our result showed that the quantity of both total and proliferative (BrdU+) IPs was increased in correlation with their periventricular mislocalization. This resulted in an overall increase in the size of the IP population along with the level of Tbr2 protein in Fln$^{cKO-NPC}$ cortices at E13.5 to E14.5 (*Figure 3J,K*, PH+, *Figure 3L*). Although increased BrdU incorporation in Tbr2+ IPs was less obvious in PH+ regions from E14.5 to E15.5 (data not shown), a 20 hr BrdU-Ki67 co-labeling analysis at this time showed that more cells quitted the cell cycle in PH+ regions (*Figure 3M*). These data, combined with a reduced number and proliferation of Pax6+ RGs in the PH+ regions, collectively suggested that the formation of PH was associated with a sequential increase in the IP pool and IPs' cell cycle exit. This local increase in both proliferation and neurogenic activities of apicalized IPs explained how periventricular neurons were produced at little expense of their neocortical counterpart.

To confirm that neurons in PH were produced by ectopic Tbr2+ IPs directly, we performed a pseudo-fate mapping experiment and crossed the Fln$^{cKO-NPC}$ mice with the *Tbr2*$^{eGFP}$ line in which cells expressing *Tbr2* are marked by GFP (*Kowalczyk et al., 2009*). Because the GFP protein has a relatively long half-life, it also marks some daughter neurons newly generated by Tbr2+ IPs. When PH was initially detected as NeuN+ clusters near the ventricle at birth, we found many of the ectopic NeuN+ neurons in PH nodules also showed GFP, indicating that PH neurons were daughters of Tbr2+ IPs (*Figure 3—figure supplement 3*).

Although a majority of *Flna*$^{Ky}$ mutants die of hemorrhage before E15, a few that survived to E15.5 showed similar phenotypes to those Flna$^{cKO-NPC}$;*Flnb*$^{+/-}$ and Flna$^{cKO-NPC}$;*Flnb*$^{-/-}$ mutants (*Figure 3—figure supplement 2F,G*), indicating Flna is the major player though it is functionally redundant with Flnb in cortical NPCs. Together, our experimental evidence demonstrates compellingly that loss of Flna in NPCs results in the mislocalization of IPs from the SVZ to the periventricular space where they gain proliferation and neurogenesis potential.

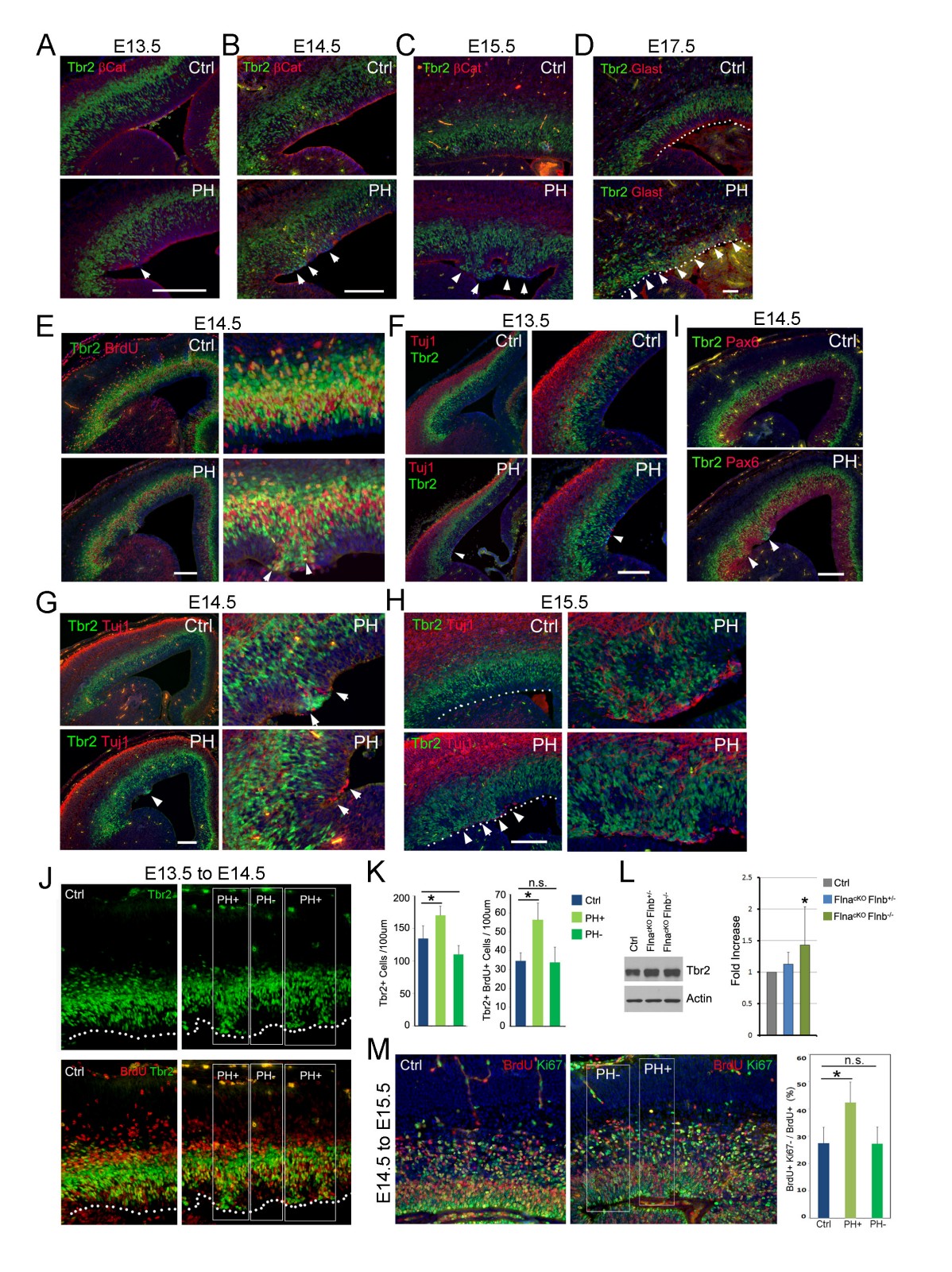

**Figure 3.** Coincidence of sporadic loss of RG adhesion with increased periventricular mislocalization, proliferation, and neurogenesis of IPs in Fln^cKO-NPC cortices. (**A–D**) Tbr2 (green) and βCatenin or Glast (red) double immunostained cortical sections, showing the progressive yet sporadic loss of AJs and the apicalization of IPs in PH brains (indicated by arrows) from E13.5 to E17.5. The ventricular surface in D is marked by dotted lines. (**E**) Tbr2 and BrdU double immunostained cortical sections from BrdU transiently labeled embryos at E14.5. Arrows indicate proliferating Tbr2+ IPs at the ventricular

*Figure 3 continued on next page*

Figure 3 continued

surface. (F–H) Tbr2 (green) and Tuj1 (red) double immunostained cortical sections, showing the coincidence of IP apicalization of IPs and periventricular neuron production (indicated by arrows) in PH brains from E13.5 to E15.5. The ventricular surface in H is marked by dotted lines. (I) Tbr2 (green) and Pax6 (red) double immunostained cortical sections, showing regional disorganization of RGs and IPs in PH brain at E14.5. (J, K) Quantitative analysis of the number and transient amplification of IPs in control and Fln$^{cKO-NPC}$ cortices. Embryos were pulse labeled with BrdU at E13.5 and analyzed for cells with Tbr2 and BrdU double immunolabeling at E14.5. Cells in regions showing apicalized IPs (PH+) were scored separately from adjacent regions where IPs retain the normal basal SVZ localization (PH-). Cell numbers were normalized by the length of ventricular surface (of 100 um). Data were acquired from three independent litters and are presented as Mean + SD; *: significantly increased Tbr2+ and Tbr2+;BrdU+ cells are detected specifically in PH+ regions (p<0.001). (L) Immunoblotting analysis of cortical Tbr2 protein levels. Data were acquired from more than three independent litters and are presented as Mean + SD; *: a significant increase was detected in the Flna$^{cKO}$;Flnb$^{-/-}$ cortices (p<0.03). (M) Cell cycle exit profiles of control and Flna$^{cKO}$; Flnb$^{-/-}$ cortical neural progenitors. Embryos were pulse labeled with BrdU at E14.5 and analyzed at E15.5 by double immunostaining of BrdU (red) and Ki67 (green). Cells that exit the cell cycle were identified as those that were positive for BrdU but negative for Ki67. Regions with apicalized BrdU+ cell (PH+) were scored separately from those with well-preserved progenitor domains in the same mutant brain (PH-). Data were acquired from three independent litters and are presented as Mean + SD; *: compared to controls, the fraction of cells that quitted the cell cycle is significantly higher in PH + regions (p<0.001). Nuclei DNA was stained with Hoechst 33,342 and shown in blue in all fluorescent images. Bars: 100 μm.

The following figure supplements are available for figure 3:

**Figure supplement 1.** Effective Flna protein abrogation from NPCs lineages and lack of cortical structural impairment in all Flna-Flnb compound mutant brains at E12.5.

**Figure supplement 2.** Ectopic division of IPs and presence of periventricular neurons without neocortical impairment or apoptosis in Fln$^{cKO-NPC}$ and Flna null (Flna$^{Ky}$) mutants.

**Figure supplement 3.** Immunohistological images of a Tbr2$^{eGFP}$ pseudo-lineage tracing study.

## Sporadic but comprehensive loss of epithelial-like features of mutant RGs

The periventricular IP mislocalization and neurogenesis was a sporadic event, but it was coincided exclusively with loss of AJ in some RGs of Fln$^{cKO-NPC}$ embryos. Furthermore, we found the sporadic RG junctional defect involved deconstruction of multiple epithelial junctional complexes beyond AJs. Loss of tight junction (TJ) protein ZO-1 and apical junction protein Pals1 in Fln$^{cKO-NPC}$ mutants was observed along with AJ disruption (*Figure 4A–D*, arrows). The broad disassembly of cell-cell junction complexes led to the dissolution of the apical F-actin belt yet remained confined and made no detectable impairment in neighboring cortical regions (*Figure 4E,F*, arrows). The profound defect of mutant RGs disrupted their apical-basally polarity as revealed by disorganized basal bodies and radial glial fibers (*Figure 4G–K*, *Figure 4—figure supplement 1*). The RGs in the affected regions appeared to lose the apical constriction that maintains the integrity of their counterparts in the adjacent unaffected regions. Such imbalance in in apical constriction led to the invagination of regions where RGs were intact and sometimes result in the formation of rosette-like structures (*Figure 4G– K*, arrows and asterisks). As the VZ invagination occurred in the unaffected RGs, it suggested that it was a secondary effect caused by altered structure of those adjacent defective RGs as well as the increased proliferation and neurogenesis of neighboring IPs. These results suggest that filamin is primarily essential in RGs through which it maintains the VZ-SVZ compartmentalization by assuring RGs' polarity and apical adhesion. Loss of filamin thereby results in the mislocalization of IPs from the normal SVZ to VZ.

Although filamins are known as a family of major actin binding proteins that cross-links and stabilizes F-actin, structural impairment in Fln$^{cKO-NPC}$ VZ involved only a subset of a cells; analysis of a large cohort of Fln$^{cKO-NPC}$ embryos throughout the entire course of cortical development failed to detect a CP defect. Despite the severe disruption of RGs in apical domains, basal radial glial fibers in the affected regions were largely intact (*Figure 4—figure supplement 1*). Therefore, the phenotype of loss of filamin only involves selected RGs and is confined in the apical domain. This is fundamentally different from previously reported cortical defects resulting from loss of structural maintenance by mutations of epithelial junctional molecules, cytoskeletal proteins, or RhoA, Rac, and Cdc42. Our data argue against a simple role of filamin in actin-dependent structural

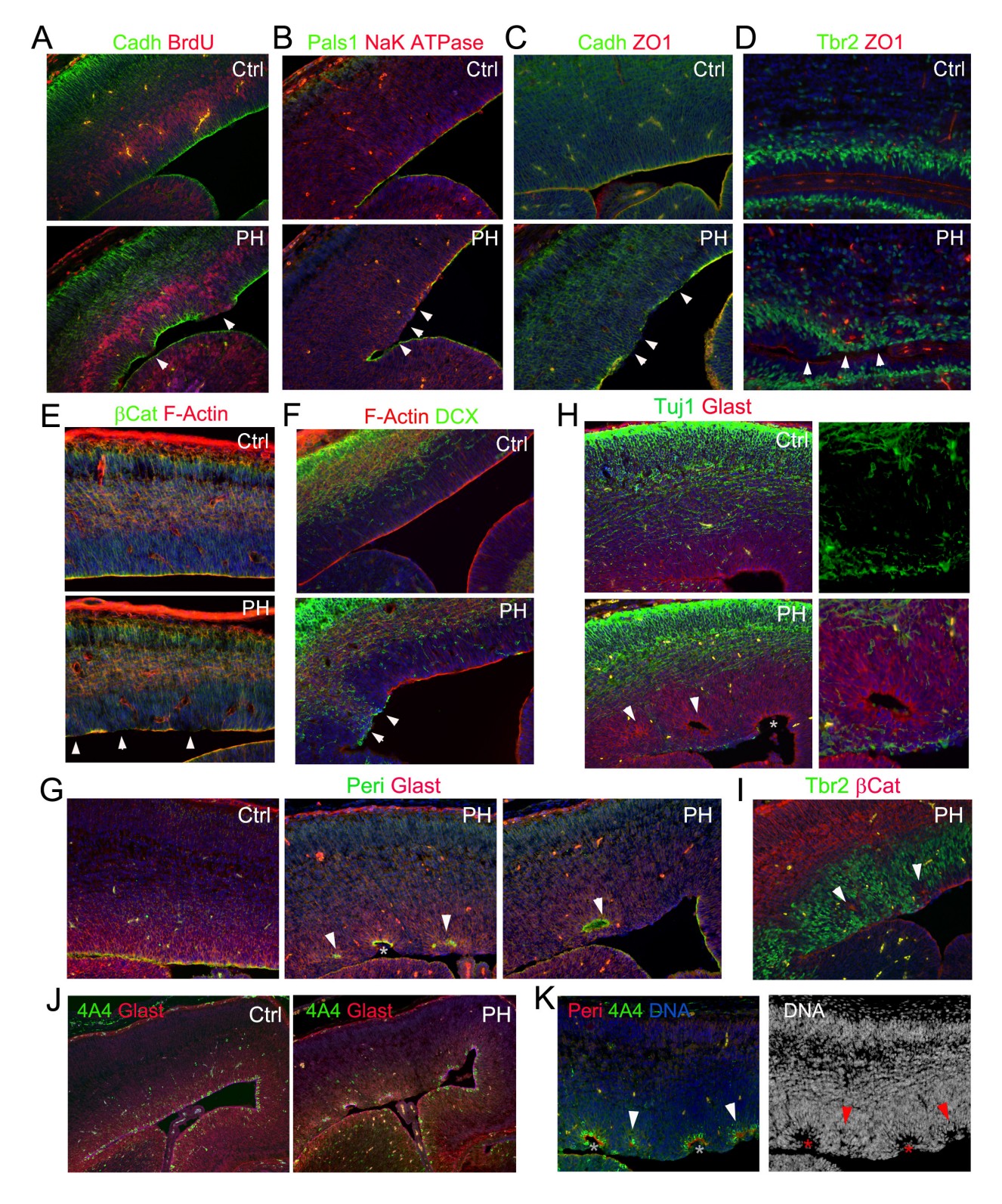

**Figure 4.** Loss of multiple epithelial features of filamin mutant RGs. (A) panCadherin (green) and BrdU (red) double immunostained cortical section of embryos fixed 30 min after a BrdU pulse at E14.5. (B) Apical junction complex marker Pals1 (green) and basal lateral marker NaK ATPase (red) double immunostained cortical sections at E14.5. (C) panCadherin (green) and tight junction marker ZO1 (red) double immunostained cortical sections at E14.5. (D) Tbr2 (green) and ZO1 (red) double immunostained cortical sections at E17.5. (E) βCatenin antibody (green) and F-actin (AlexaFluor 568 Phalloidin,

*Figure 4 continued on next page*

*Figure 4 continued*

red) stained cortical sections at E14.5. (F) The newborn neuron marker DCX (green) and F-actin (AlexaFluor 568 Phalloidin, red) double stained cortical sections at E14.5. (G) Pericentrin (marks the basal bodies at the apical side, green) and Glast (labels the basal processes of RGs, red) double immunostained cortical sections at E15.5. Note that the comprehensive loss of RGs' polarity and junctional complexescortices was accompanied by reduced mechanical strength, resulting in protrusions wrapping around the adjacent RGs and the forming rosette-like structures in the PH brain (arrows). (H) Glast (red) and Tuj1 (green) double immunostained cortical sections at E15.5, showing rosettes were formed around ectopically produced periventricular neurons in the PH brains. (I) β—Catenin (red) and Tbr2 (green) double immunostained cortical sections at E15.5, showing that rosettes were formed around mislocalized IPs in the PH brains. (J, K) Triple or double immunostaining with antibodies to Glast, pericentrin, and metaphase progenitor marker phospho-vimentin 4A4 (colored as indicated on the image), which showed that loss of RG adhesion and polarity was associated with reduced RG mitosis at E15.5 in PH brains. In (A)-(F), regions with defective RGC-RGC apical adhesion, Tbr2+ IP apicalization and ectopic neurogenesis in Fln$^{cKO-NPC}$ cortices are indicated be arrows. In (H)-(K), fully formed rosettes are indicated by arrows; partially formed rosettes are indicated by asterisks. Nuclei DNA was stained with Hoechst 33,342 and shown in blue in all fluorescence images.

The following figure supplement is available for figure 4:

**Figure supplement 1.** Unremarkable change in the basal processes of RGs in regions affected by loss of filamin.

maintenance, and they instead suggest that the sporadic but comprehensive loss of epithelial-like characteristics as well as *Pax6* expression in selected mutant RGs represents a transition in their identity.

## Sustained EMT of RGs surrounding mislocalized IPs

The stochastic loss of epithelial polarity, multiple cell-cell junctions, and reduced mechanical strength in RGs of Fln$^{cKO-NPC}$ embryos altogether resemble epithelial-mesenchymal transition (EMT), a multi-step and reversible processes that cause profound changes in a cell's structure and identity (*Thiery et al., 2009*). To test if the RG defects in Fln$^{cKO-NPC}$ VZ were associated with abnormal EMT, we first examined EMT promoting transcription factors Snail and Twist by in situ hybridization (ISH). After detecting *Snail1* and *Twist1* mRNAs in Fln$^{cKO-NPC}$ embryos and their control littermates, we overlaid ISH images with IH images of β-Catenin, Cadherin, Tbr2 or Tuj1 to identify the relationship between cells with high *Snail* or *Twist* and those with aberrant adhesion or ectopic localization. At E13.5 to E14, a small number of cells in the VZ of the Flna$^{cKO;}$Flnb$^{-/-}$ cortex was found to express a higher level of *Snail1* (*Figure 5A–D*). We showed that cells with elevated *Snail* were precisely those that lost AJs (*Figure 5A,B*) and these *Snail*$^{high}$ cells were also the nearest neighbors of the ectopic IPs and neurons (*Figure 5C,D*). Similarly, increased expression of *Twist1* was also found to correlate with periventricular IP mislocalization and ectopic neuron generation in Fln$^{cKO-NPC}$ cortices (*Figure 5—figure supplement 1A*). The enhanced EMT-like change in the mutant RGs was further demonstrated by increased mesenchymal phenotypes indicated by an excess amount of Fibronectin (FN) in regions with weakened RG adhesion (*Figure 5E*). Moreover, increased hyaluronic acid (HA), a mesenchymal extracellular matrix (ECM) glycoprotein associated with rapid tissue remodeling, was shown by the mutant and coincided with disassembled AJs and ectopic neurons (*Figure 5F*, *Figure 5—figure supplement 1B*). Collectively, the concurrent loss of adhesion with gained mesenchymal ECM and EMT promoting transcription factors confirmed an EMT-like change in Fln$^{cKO-NPC}$ RGs, which can lead to profound alteration of gene expression and local ECM composition and create a brand new microenvironment to potentiate the transient amplification of the local IPs as well as their capacity in generating neurons.

To test the direct participation of FLNA in EMT, we studied the effect of mutant forms of filamin A (FLNA) on EMT induced by TGFβ in MDCK cells. Both filamin A and filamin B are composed of an N terminal actin binding domain (ABD) followed by 24 Ig-like repeats, the last of which mediates their homo and hetero-dimerization. The C-terminal domain of filamins is also unique in mediating numerous molecular interactions with cell signaling molecules (*Feng and Walsh, 2004*; *Zhou et al., 2010*). Thus, retroviruses producing an actin binding domain deleted FLNA (FLNA-ΔABD) (*Kainulainen et al., 2002*) and the FLNA C-terminal molecular interaction domain (FLNA-C, encompassing amino acids 2168–2647) were used to interrupt FLNA's role in actin remodeling and cell signaling, respectively. We found the EMT phenotypes of these two mutant lines were remarkably different from vector infected controls. While the FLNA-ΔABD line showed an enhanced response to

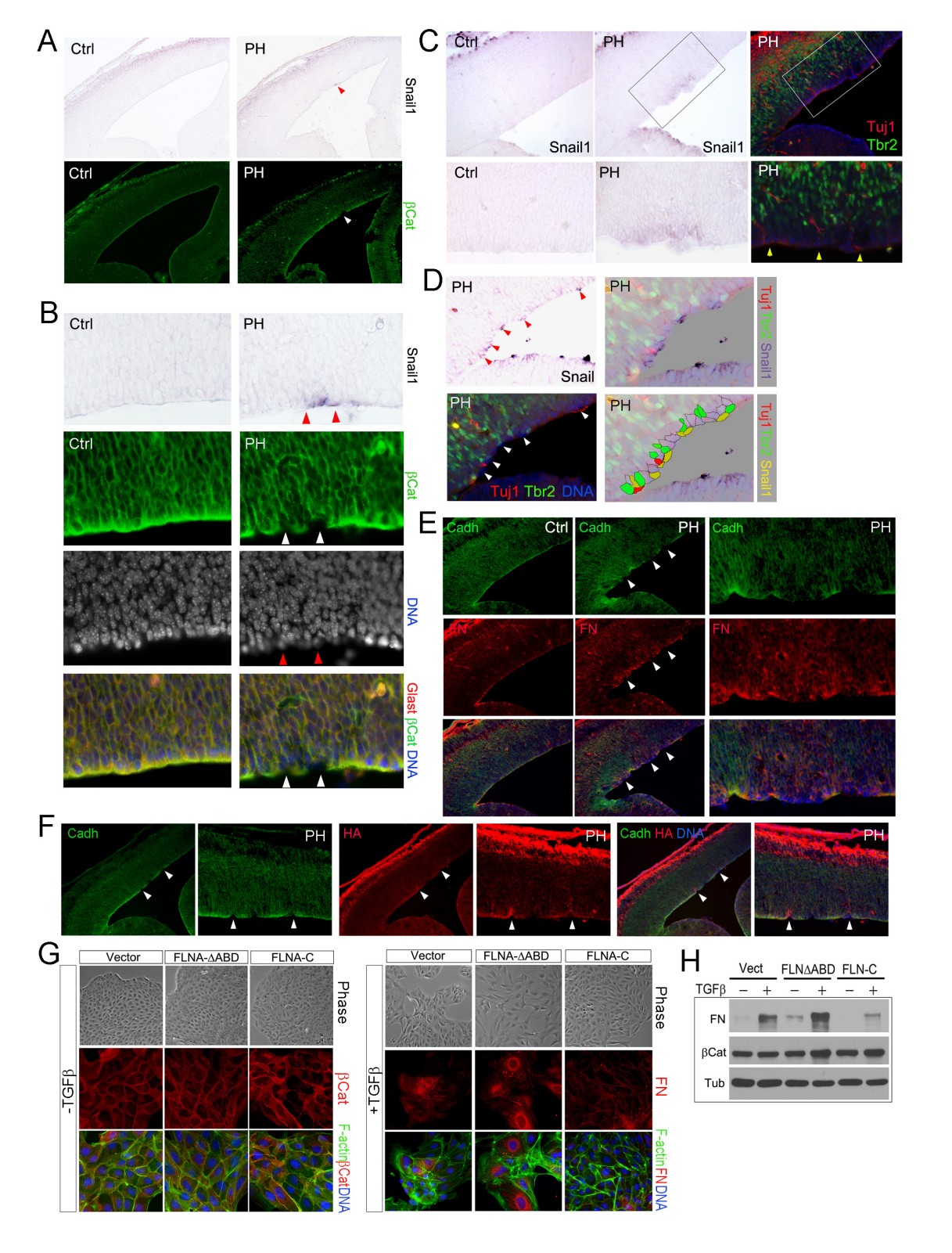

**Figure 5.** Sustained EMT in RGs of Fln^cKO-NPC cortices and a direct role of FLNA in EMT regulation. (**A, B**) In situ hybridization (ISH) and immunohistological analyses show the correlation of increased *Snail1* expression with loss of apical β-Catenin in the cortical VZ of Fln^cKO-NPC embryos at E13.5. Tissue sections were subjected to ISH first; they were then co-immunostained by antibodies to β-Catenin (green) and Glast (red). Cells with high *Snail*1 and loss of apical β-Catenin are indicated by red and white arrows, respectively. (**C, D**) In situ hybridization and immunohistological analyses

*Figure 5 continued on next page*

Figure 5 continued

show the coincidence of cells with elevated *Snail1*, apicalized IPs, and ectopic neurons in filamin mutant VZ at E13.5. Tissue sections were subjected to ISH; photographed, then co- immunostained by antibodies to Tbr2 (green) and Tuj1 (red). Arrows indicate ectopic neurons along the ventricular surface. Overlaid images of *Snail1* in situ hybridization and Tbr2, Tuj1 immunostaining are included to reveal the spatial relationship between *Snail1*[high] RGs, Tbr2+ IPs, and Tuj1+ neurons. (E) panCadherin (green) and Fibronectin (FN, red) double immunostained cortical sections at E14.5. Higher magnification images of the mutant VZ are included. (F) The coincidence of apical panCadherin (green) destabilization with increased hyaluronic acid (HA, red) in Fln[cKO-NPC] cortices at E13.5. Regions with HA elevation were indicated by arrows. (G) Phase contrast and immunofluorescence images of MDCK cell lines stably transfected by vector, FLNA-ΔABD or FLNA-C, and cultured in the absence or presence of TGFβ for 6 days. β-Catenin (red) and F-actin (green) are used to view epithelial junctions before TGFβ induction, whereas Fibronectin (FN, red) and F-actin (green) are used to indicate mesenchymal cell like features after TGFβ induction. (H) Immunoblotting analysis of mesenchymal cell marker Fibronectin (FN) 6 days after TGFβ induction. Note that the total level of β-Catenin remained unaltered even though the cell-cell junctions were disintegrated following TGFβ induction. Nuclei DNA was stained with Hoechst 33,342 and shown in blue in all fluorescent images.

The following figure supplement is available for figure 5:

**Figure supplement 1.** Increased EMT in Fln[cKO-NPC] cortical tissue and a role of FLNA in EMT regulation.

TGFβ in losing epithelial morphology, reducing junctional β-catenin, and transforming into fibroblast-like cells that produce higher levels of Fibronectin (FN) and HA, the FLNA-C cells showed impeded EMT with persistent epithelial morphology and decreased FN production (*Figure 5G,H*; *Figure 5—figure supplement 1C–D*). These results not only indicate that FLNA is an EMT regulator and likely plays a role in integrating EMT inducing signals with actin remodeling but also suggest that *FLNA* mutations could lead to aberrant EMT in epithelial cells of multiple tissues.

## The association of PH-genesis with increased angiogenesis

Despite the direct involvement of filamin in EMT, the EMT-like phenotype in RGs of Fln[cKO-NPC] embryos was sporadic. This suggested the essential contribution of a non-cell autonomous factor to sustain EMT promoting changes in the mutant RGs. We therefore looked for other phenotypes in Fln[cKO-NPC] cortices and found that the sporadic RG and IP defects coincided with altered angiogenesis. During cortical development, blood vessels and NPCs develop congruently and vasculogenesis not only occurs in parallel to neurogenesis but also is tightly associated with the proliferative activity of IPs (*Javaherian and Kriegstein, 2009*; *Stubbs et al., 2009*; *Vasudevan and Bhide, 2008*). Additional to previous reports, we found angiogenesis sprouting from pial arteries to ventricles preceded IP generation and proliferation. While blood vessels penetrated to the VZ by E12.5, they remained largely segregated from the IPs. By E13.5 to E14.5 both blood vessels and IPs showed a substantial increase in abundance, and IPs started to intermingle with blood vessels and divide more robustly (*Figure 6—figure supplement 1*). At this stage blood vessels in the cortical VZ of Fln[cKO-NPC] embryos were shown to be irregular in terms of abundance, distribution and structure compared to controls (*Figure 6A–E*; *Figure 6—figure supplement 1A–D*). Quantitative analysis of vessel density indicated an overall increase in the vascular bed of Fln[cKO-NPC] cortices (*Figure 6F*). Although vascular endothelium cells (ECs) were intact and pericyte coverage was detectable, nascent vessels in the mutant were grossly coarse with larger caliber (*Figure 6G*). While blood vessels in control cortices form a plexus in the CP and SVZ, many vessels in Fln[cKO-NPC] cortices sprouted deeply to the periventricular space where they coincided with destructed RG adhesions (*Figure 6A, B, H*; *Figure 6—figure supplement 2A,C*). Remarkably, we found that the defective RGs as well as ectopic IPs and neurons in Fln[cKO-NPC] cortices inevitably commingled with aberrant vessels (*Figure 6A–E*, *Figure 6—figure supplement 2A–D*). Cerebral hemorrhage and enlarged blood vessels were also seen in Fln[cKO-NPC] postnatal brains and found inside and/or adjacent to periventricular heterotopic neuronal nodules (*Figure 6I*, *Figure 6—figure supplement 2E*). These findings suggested strongly that the sporadic phenotype in selected RGs and IPs of Fln[cKO-NPC] cortices was associated with altered vascular structure and function.

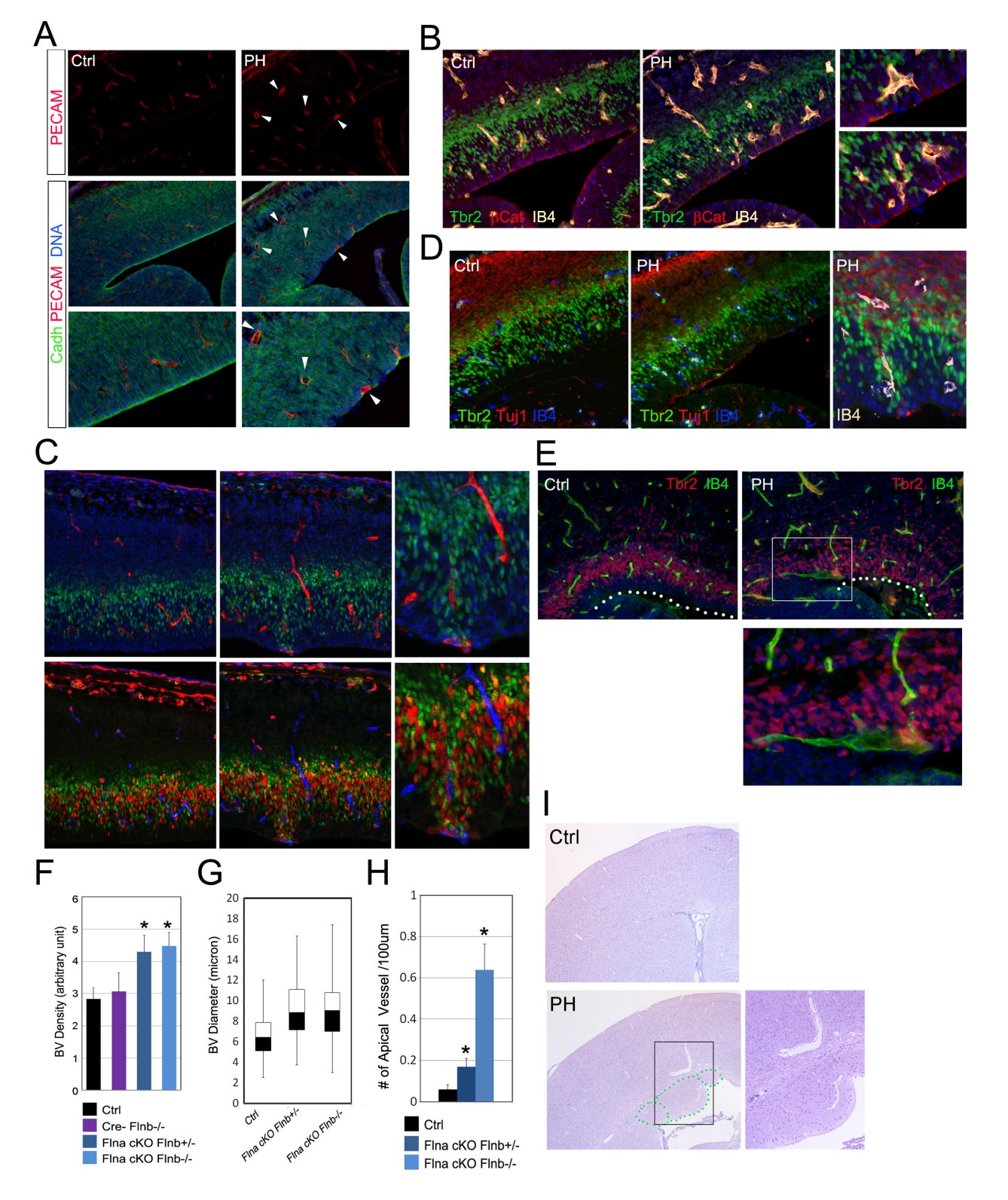

**Figure 6.** Coincidence of neural and vascular phenotypes in Fln[cKO-NPC] cortices. (**A**) PECAM (red) and pan-Cadherin (green) double immunostained cortical sections at E14.5. Arrows indicate enlarged periventicular blood vessels in Fln[cKO-NPC] cortices. (**B**) β-Catenin (red), Tbr2 (green), and Biotin-IB4 (yellow) triple stained cortical sections of control and PH at E14.5. Higher magnification views show the intermingling of periventicular blood vessels with defective RGs and IPs in the mutant. (**C**) BrdU (red), Tbr2 (green), and Biotin-IB4 (red or blue as indicated) triple stained cortical sections of control

*Figure 6 continued on next page*

*Figure 6 continued*

and PH embryos at E15.5. Embryos were fixed 30 min after a BrdU pulse. Higher magnification views reveal the coincidence of periventricular blood vessels and proliferative IPs. (**D**) Tuj1 (red), Tbr2 (green), and Biotin-IB4 (blue or white as labeled) triple immunostained cortical sections at E14.5. A higher magnification image shows the spatial concurrence of aberrant blood vessels, apicalized IPs, and ectopic neurogenesis at the ventricular surface. (**E**) Tbr2 (red) and Biotin-IB4 (green) double stained cortical sections at E17.5. A higher magnification image shows the enlarged periventricular blood vessel in the mutant. The ventricular surface is indicated by dotted lines. (**F**) Quantification of vascular density at E14.5. Data are presented as Mean + SD from a minimum of 3 independent litters. *p<0.001. (**G**) Box and whisker plots of blood vessel diameter. Shown are medians, the 25th–75th percentiles (box), maximum and minimum values (whiskers) from 3 independent litters. (**H**) Quantification of periventricular (< 1-cell diameter from the ventricle) blood vessels per 100 μm ventricular line. Data are presented as Mean + SD from a minimum of 3 independent litters. *p<0.001. (**I**) H&E stained cortical sections from a control and a $Flna^{cKO};Flnb^{+/-}$mutant at weaning, showing the coincidence of PH and aberrantly enlarged blood vessels.

The following figure supplements are available for figure 6:

**Figure supplement 1.** Developmental cortical angiogenesis and IP-genesis.

**Figure supplement 2.** Aberrant cerebral angiogenesis with PH-genesis was observed in $Fln^{cKO-NPC}$ but not $Fln^{cKO-EC}$ mutants.

**Figure supplement 3.** Conditional abrogating Flna by *GFAP*-Cre results in PH-like phenotype.

## Escalated bi-directional growth signaling between NPCs and blood vessels

As we specifically deleted Flna from NPCs but not vascular endothelial cells (ECs) (*Figure 3—figure supplement 1A*), the vascular phenotype associated with PH must result from a NPC defect. To better demonstrate the NPC origin of the angiogenic anomaly, we abrogated Flna in ECs using the $Tie2^{Cre}$ driver but failed to detect any defect in the cerebral cortex of EC conditional Flna mutants (referred to as $Flna^{cKO-EC}$; *Figure 6—figure supplement 2F–H*). In contrast, using a GFAP-Cre driver (*Zhuo et al., 2001*) that conditionally ablated Flna from RGs after E13.5 recapitulated the PH-like phenotype of mice with Flna pan-NPC conditional knockout from early development (*Figure 6—figure supplement 3*). While these findings supported the primary requirement of Flna in RGs, they also demonstrated NPCs' or RGs' influence to cerebral angiogenesis. To ascertain that deleting Flna from NPCs resulted in vascular abnormality, we showed elevated production of several Vegfa isoforms in $Fln^{cKO-NPC}$ cortical tissue (*Figure 7A*), which were in line with increased angiogenesis in these mutants. More interestingly, elevated *Vegfa* mRNA was spatially coincided with not only the enrichment of periventricular blood vessels but also the RG aberrance (*Figure 7B*). We further showed that the increased *Vegfa* expression in the mutants was not induced by hypoxia since Hif1$\alpha$ and stress associated phospho- p38 or JNK were not elevated significantly in the cortical tissue of $Fln^{cKO-NPC}$ embryos. In contrast, we observed a specific increase in phospho-Erk, activated by growth factor signaling, in $Fln^{cKO-NPC}$ cortices (*Figure 7A*). Therefore, increased *Vegfa* expression in $Fln^{cKO-NPC}$ cortices was associated with stronger growth factor signaling and Erk activation.

The activation of Erk through the Ras-Raf-Mek-Erk kinase cascade is a common response downstream of multiple growth signals. IGF2 is known to be the major growth factor provided by the embryonic circulatory system, and was found at a high level in the PH-associated periventricular blood vessels of $Fln^{cKO-NPC}$ cortices (*Figure 7C*). As IGF2 signals through IGF1R, a receptor tyrosine kinase, we examined Igf1r and its phosphorylation in cortical tissue. Although changes in the level of Igf1r were subtle, its tyrosine phosphorylation was increased significantly in $Fln^{cKO-NPC}$ cortices (*Figure 7D,E*), supporting the notion that the stronger Erk activation and excessive angiogenesis in $Fln^{cKO-NPC}$ cortices resulted from increased Igf2 signaling. The persistent Igf1r activation also suggested a role of filamin in attenuating Igf2 signaling, and loss of this function resulted in stronger Ras-Raf-Mek-Erk activation and *Vegfa* expression in NPCs. Therefore, the higher Igf2 signals in $Fln^{cKO-NPC}$ cortices can further increase the vascular permeability via elevated Vegfa, forming a positive feedback loop that escalates Igf2 release and its constitutive signaling (*Figure 7F*). Both increased Igf1R and Erk signaling are known to induce EMT (*Gonzalez and Medici, 2014*; *Lindsey and Langhans, 2014*). Therefore, failed dampening of their signals exacerbates the EMT-like and vascular phenotypes in all-or-none coincidence in $Fln^{cKO-NPC}$ cortices.

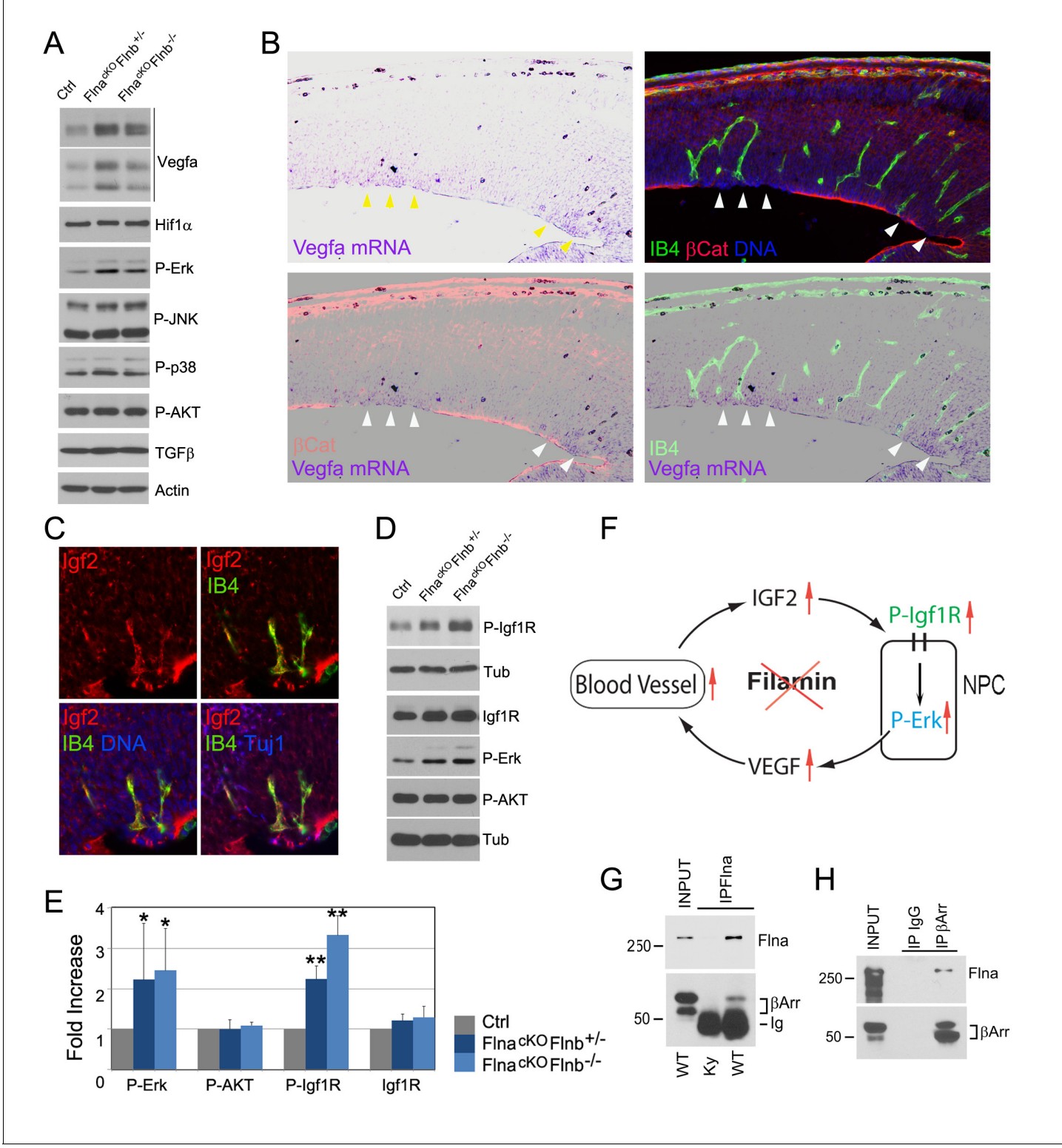

**Figure 7.** Loss of filamin function resulted in escalated Vegfa and Igf2 signaling (**A**, **D**) Immunoblotting analysis of cortical total protein extracts from E13.5 Flna^cKO^; *Flnb^+/-^*, Flna^cKO^; *Flnb^-/-^*mutants and their Cre- control littermate. (**B**) Combined *Vegfa* mRNA in situ hybridization with IB4 and β-Catenin double immunohistological analyses on cortical sections of a *Flna^flox Emx1 Cre+^;Flnb^-/-^* embryos at E14.5, revealing the spatial correlation of increased Vegfa expression with RG and antigenic anomalies (arrows). (**C**) Igf2 (red), IB4 (green), and Tuj1 (blue) triple stained cortical sections of a Fln^cKO-NPC^ embryo at E14.5, showing high Igf2 in periventricular blood vessels. (**E**) Quantitative analysis of elevated phospho-Erk (P-Erk) and phosphor-Igfr (P-Igf1R Y1161) in Fln^cKO-NPC^ cortical tissue at E13.5. Data are acquired from a minimum three litters and presented as Mean + SD; significantly increased

*Figure 7 continued on next page*

*Figure 7 continued*

phospho-Erk and phospho-Igf1R Y1161 are detected in both Flna$^{cKO}$;*Flnb$^{-/-}$* and Flna$^{cKO}$; *Flnb$^{+/-}$* cortices. **p<0.001; *p<0.05. (F) A diagram illustrating the positive feedback loop that escalated the neural progenitor and angiogenic phenotypes in Fln$^{cKO-NPC}$ cortical tissue. (G) Co-immunoprecipitation of β-arrestin with Flna. Flna immunoprecipitation was performed with cortical tissue lysates of wild type and Flna null embryos followed by immunoblotting with anti- β-arrestin. (H) Co-immunoprecipitation of Flna with β-arrestin. β-arrestin immunoprecipitation was performed with cortical tissue lysates of wild type E13.5 embryos using rabbit IgG as the negative control. An Ig light chain specific antibody was used for immunoblotting detection of Flna and β-arrestin.

FLNA is a multifaceted molecule with tissue context-dependent functions mediated through its molecular interactions. Out of numerous FLNA interacting proteins reported previously, β-arrestin plays an important role in both desensitizing IGF1R and controlling ERK activation (*Girnita et al., 2007*; *Kovacs et al., 2009*; *Lin et al., 1998*; *Scott et al., 2006*; *Zheng et al., 2012*). β-arrestin is highly enriched in the developing cerebral cortex. To determine whether failed attenuation of mitogenic signals in Fln$^{cKO-NPC}$ RGs was due to an impaired β-arrestin function, we examined the Flna-β-arrestin interaction in embryonic cortical tissue by co-immunoprecipitation. Not only was β-arrestin detected in Flna immunoprecipitates, we also showed that Flna was specifically co-precipitated with immunoprecipitated β-arrestin (*Figure 7F,G*), indicating Flna is a partner of β-arrestin in cortical NPCs. Therefore, our data together suggest that the interaction and corroboration of filamin and β-arrestin play an essential role in dampening the excessive Igf2 signals and maintaining a homeostatic communication between NPCs and blood vessels in the developing cerebral cortex. Loss of filamin impairs β-arrestin's role in down-regulating Igf2-Igfr signaling, which augments both vascular and NPC growth and activities.

## Discussion

In summary, data from this study demonstrate that the neurogenic potential of cortical IPs can be enhanced by microenvironmental alterations in the filamin mutant cerebrum. Factors contributing non-cell-autonomously to the net gain of ectopic neurons produced by IPs include ECM changes from an EMT-like event as well as increased local angiogenesis and vascular activity. Our data indicate that the essential requirement of Flna in cerebral cortical development is to assure the homeostatic interaction among RGs, IPs, and nascent vasculature. By interacting with β-arrestin, Flna blocks the excessive communication between blood vessels and NPCs through attenuating growth signals. Therefore, loss of Flna results in constitutive activation of Igf1R to Erk signaling and increased production of Vegfa, which exacerbate both NPC and angiogenic responses in an escalating manner and cumulate to sustained EMT of RGs. These together resulted in altered niche of IPs and boosted their neurogenesis at locations where RG and vascular changes were irreversible.

One of the most surprising results of this study is that we did not detect compromised production and migration of neurons destined for the neocortex despite the remarkable destruction to the VZ and SVZ after E14.5 in PH-forming Fln$^{cKO-NPC}$ cortices. The lack of impairment of neocortical neurons with the presence of PH is in line with a recent finding from a fate mapping study which suggests that NPCs destined to generate neurons in upper cortical layers become fate restricted in early corticogenesis (*Franco et al., 2012*). It is also likely that RGs stop making cortical neurons directly after E13.5 so that the integrity of the apical compartment of the cortical VZ and the apical RG-RG junctions become dispensable for the generation and migration of neocortical neurons, and the bulk of neurons that migrate into the CP after E13.5 were from basally localized IPs. Overall, our findings support the notion that the neurogenic potential of NPCs can be manipulated and enhanced in vivo and there is likely a surplus of IP population that remains sensitive to microenvironmental inputs and is responsible for generating periventricular neurons in filamin mutants. It is especially appealing that the surplus NPCs and their neurogenic plasticity in the cerebrum of Fln$^{cKO-NPC}$ mice occur at the level of IPs, since IPs are neuronal fate-restricted progenitors that only give rise to cortical neurons. The size and diversity of the IP pool increase with the neocortical evolution. IPs are believed to be responsible for the generation of the majority of the cortical projection/excitatory neurons and play an ongoing role in postnatal neurogenesis (*Hodge et al., 2008*; *Kowalczyk et al., 2009*; *Zhao et al., 2008*). Therefore, factors that may alter the plasticity of IPs would specifically and

directly boost neuron production. Identifying these factors may open a portal for inducing the specific generation of neurons in cerebral tissue without affecting cells of the other lineages. This could potentially increase the likelihood of neural stem/progenitor cell mediated therapeutic intervention.

Our experimental findings demonstrate that the proliferation and neurogenic plasticity of IPs is highly sensitive to local tissue context, which includes ECM composition and blood supply. Increased crosstalk among vascular ECs, RGs and IPs appears essential for a positive reinforcement that creates a favorable microenvironment for IPs' transient amplification and neurogenesis. One of the ways to establish such a microenvironment may be through blocking FLNA and its associated molecules' specific function in dampening the growth signaling in NPCs and their neighboring cells, since this could both augment the impact of growth factors from the blood circulation and potentiate EMT-like changes to alter the local ECM and cytomechanical properties. As a fundamental and multi-step process for tissue morphogenesis, EMT involves the synergistic action of a hierarchical molecular network and is estimated to alter the expression of over a thousand genes. The ECM molecules produced by the resulting mesenchymal-like cells can create a brand-new microenvironment for both NPCs and ECs. Although PH represents an abnormal condition of increased IP neurogenesis, further defining the molecular signature of the periventricular niche responsible for increased neurogenesis in the FLNA mutant brain may provide key insights into the cell molecular tools required for boosting the neurogenic potential of IPs in cerebral tissue.

Although FLNA is best known as an actin binding protein, our result challenges its simple mechanical function but instead highlights a novel mechanism in maintaining tissue homeostasis by acting as a cell signaling modulator. During cortical development, FLNA ensures proper neurovascular congruency by regulating bi-directional signaling and intercellular interaction between NPCs and vascular cells. Through collaborating with β-arrestin, FLNA serves as a cytoplasmic molecular break to attenuate the excessive growth signals and maintain the homeostasis at both cell and tissue levels. While our study reveals escalated Vegfa and Igf2 activities, imbalances in many other molecules and pathways could also be involved in promoting the NPC and vascular phenotypes in filamin deficient cortices. Besides Igf1R, β-arrestin can mediate the turnover and desensitization of many plasma membrane receptors through clathrin mediated endocytosis or by recruiting ubiquitin ligases (*Lin et al., 2008*). FLNA has also been reported to interact with numerous membrane receptors and cell signaling adaptors (*Feng and Walsh, 2004*). The extraordinarily diverse array of molecular dynamics governed by neurovascular communication and EMT-like change of filamin mutant RGs, in fact, suggest that loss of filamin results in a complex alteration in the interplay of multiple cell types and molecules, which would be extremely interesting to elucidate in future studies.

## Materials and methods

### Mouse strains

*Flna* knockout (*Flna*^K/w or *Flna*^K/y), floxed (*Flna*^flox/y or Flna^cKO) and *Flnb* knockout (*Flnb*^-/-) mice have been described previously (*Feng et al., 2006*; *Lu et al., 2007*). The *Emx1*^Cre, *Tie2*^Cre, and *GFAP*^Cre mice were obtained from JaxMice (stock # 005628, #008863, and #004600, respectively). The Tg (*Eomes*-EGFP) Tbr2^eGFP mice were obtained from GENSAT (Item #011151-UCD-SPERM). The *Flna* and *Flnb* double mutant mice were generated by standard genetic crosses. All mice used for this study were housed and bred according to the guidelines approved by the IACUC committee of Northwestern University. For timed matings, the day of vaginal plug was considered E0.5.

### Immunohistology and antibodies

Immunohistology studies were carried out as described (*Pawlisz et al., 2008*) on 12 μm frozen or 5 μm paraffin sections. The following antibodies were used: FLNA, Pals 1, Igf2 (Epitomics); Cux1, DCX (Santa Cruz); Pax6, Neurofilament, ZO1, NaK ATPase (Developmental Study Hybridoma Bank); BrdU B44, βCatenin, E-Cadherin, N-Cadherin, VE-Cadherin, PECAM, Fibronectin, beta Integrin (BD Biosciences); Tuj1, Pericentrin (Covance); BrdU BU1/75, Foxp1, Foxp2, SATB2, Ctip2, Glast, (Abcam, Cambridge, MA); GFAP (Dako); NeuN, Tbr2, Tbr1, HABP (Millipore); panCadherin, GFP (Life Technologies); MAP2, Biotin-IB4 (Sigma); PDGFRβ (eBioscience); Alexa Fluor 488 Phalloidin and Alexa Fluor 546 Phalloidin and fluorescence conjugated secondary antibodies were from Life

Technologies. All experiments were repeated with at least three independent litters and representative images are shown.

## Plasmids, cell culture, FLNA retrovirus, stable cell lines and EMT analysis

293FT and MDCK cells were from ATCC (PTA-5077 and PTA-6500, respectively) and grown in DMEM supplemented with 10% FBS and 50 µg/ml gentamicin. The cultures were maintained and periodically tested to ensure mycoplasma-free. MDCK culture-based EMT study was performed essentially as described previously (*Medici et al., 2006*; *Wyatt et al., 2007*) using two FLNA constructs cloned into pBabe-Puro for the generation of stable MDCK lines.

To construct pBabe-mycFLNA-C, cDNA encoding amino acids 2168 to 2647 of FLNA (Ig repeats18 through 24) was amplified by PCR using primers ATG GAA CAA AAGTTGATT TCTGAA-GAAGATTT GGAAATTAGCATCCAGG and GT GACCAGACTCAGGGCACCAC, and cloned into the pCRII vector using the TA cloning Kit (Invitrogen). After sequence confirmation, the mycFLNA-C was released from the TA clone by restriction digestion with Sal I and Not I, then sub-cloned into pBabe-Puro between Bam HI and Sal I with a Not I-BamH1 linker. The pBabe-myc FLNA-ΔABD was constructed in two steps: An EcoR1 – Xba I fragment of the C-terminal end of the pcDNA3-FLNA-Δ ABD cDNA (*Kainulainen et al., 2002*) was first cloned into pBabe-Puro between EcoR I and Sal I using an Xba I-Sal I linker. Then the N-terminal Hind III-EcoR I fragment of the pcDNA 3-mycFLNA-Δ ABD was inserted between Bam HI and EcoR I with a Bam HI-Hind III linker.

Retroviruses expressing mycFLNA-C and mycFLNA-ΔABD were packaged in 293FT cells by co-transfection with VSV-G and a plasmid expressing Gag-Pol using Lipofectamine (Life Technologies) essentially as described (*Morgenstern and Land, 1990*). The culture supernatant was harvested in 48 hr, filtered and used to infect MDCK cells with 2 µg/ml polybrene. Stably transfected MDCK cell lines were generated by infecting with FLNA-ΔABD or FLNA-C retroviruses for 2 days followed by selection of resistance to 2 µg/ml puromycine. Single clones of FLNA-ΔABD and FLNA-C were isolated, verified by myc- immunofluorescence or western blotting before being used for analysis. A pool of pBabe-Puro vector infected MDCK clones were used as the negative control.

For EMT induction, recombinant TGF-β (Peprotech) was added to the MDCK culture medium at a concentration of 5 ng/ml. The culture was maintained at 30–90% confluency in the presence (or absence) of TGF-β for 3 to 6 days. Immunofluorescence and western blotting were performed as previously described (*Pawlisz and Feng, 2011*).

## In situ hybridization

In situ hybridization analyses with mouse tissues were carried out as described previously (*Pawlisz et al., 2008*). An mRNA probe for *Snail1* was generated using T3 RNA polymerase from the murine *Snail1* IMAGE clone 5121591 linearized by EcoN1. An mRNA probe for *Twist* 1 was generated by T7 polymerase from the murine *Twist 1* IMAGE clone 4935230 linearized by Sph I. The *Vegfa* mRNA probe was generated by constructing a plasmid through PCR amplifying the mouse *Vegfa* cDNA including 3' UTR with ATCTGTGTTTCCAATCTCTCTC and ATGTACTACGGAATATCTCGG primers and cloning into the pCRII vector. After sequence confirmation, the plasmid was linearized by Bam H1 and transcribed with T7 RNA polymerase.

## Immunoblotting and Immunoprecipitation

Immunoblotting of total cortical proteins was performed using extracts of hot SDS PAGE sample buffer containing 2% SDS from filamin mutants, wild type or Cre- control littermates at E13.5. Antibodies used include Tbr2 (Abcam), Pax6 (DSHB), TGFβ (Abcam); Hif1α, VEGF (Novus Biologicals); phospho-Erk, p38, JNK and AKT (T308 and S473), β-arrestin (Cell Signaling), Igf1Rb, P-Igf1R (Y1161) (Santa Cruz), α-Tubulin, Actin (Sigma). Immunoprecipitation was performed with cortical lysates prepared from E13.5 embryos in a buffer containing 100 mM NaCl, 20 mM KCl, 25 mM HEPES, pH 7.6, 2 mM EDTA, 0.2% Brij-96, 25 µg/ml Leupeptin, 25 µg/ml Pepstatin A, 10 mM Benzamidine, and 1 mM PMSF. To avoid overlapping signals between Ig heavy chain and β-arrestin, a HRP conjugated Ig light chain specific secondary antibody was used for immunodetection. The experiments were repeated with at least three independent litters, immunosignals were quantified by Image J, and representative images are shown.

## Image analysis

Quantitative analysis of immunohistological images was performed using the Image J software with spatially matched brain sections from mutants and wild type or Cre- control littermates. Cell counts were obtained from the entire the cortical wall and divided by the length of ventricular surface. Vascular density was estimated on IB4 stained cortical sections using the grid overlay plugin of Image J by scoring the intersection between blood vessels and the counting grid.

## Statistical analysis

Student's t tests (two groups) or ANOVA (groups $\geq$ 3) were performed to determine significant differences between different genotypes in most experiments. Chi-square test was used for BrdU birth-dating studies to determine categorical cell fractions. All data were presented with mean values and standard deviations. Differences were considered significant with a p value < 0.05.

## Acknowledgements

The authors wish to thank Drs. Anjen Chenn (Univ Illinois at Chicago) and Chonghui Cheng (Northwestern) for discussion and suggestions; Renjie Li for excellent technical help with mouse histology; Dr. Volney Sheen (BIDMC, Harvard Medical School) for sharing the *Flnb* knockout mice. This work is supported by R01NS087575 and the Brain Research Foundation.

## Additional information

### Funding

| Funder | Grant reference number | Author |
| --- | --- | --- |
| National Institute of Neurological Disorders and Stroke | R01NS087575 | Yuanyi Feng |
| Brain Research Foundation | | Yuanyi Feng |

The funders had no role in study design, data collection and interpretation, or the decision to submit the work for publication.

### Author contributions

SLH, AAL, Acquisition of data, Analysis and interpretation of data, Drafting or revising the article; YG, Acquisition of data, Analysis and interpretation of data; YF, Conception and design, Acquisition of data, Analysis and interpretation of data, Drafting or revising the article

### Author ORCIDs

Yuanyi Feng, http://orcid.org/0000-0003-2793-3962

### Ethics

Animal experimentation: This study was performed in strict accordance with the recommendations in the Guide for the Care and Use of Laboratory Animals of the National Institutes of Health. All of the experimental mice were handled according to the animal protocol (#IS0001492) approved by institutional animal care and use committee (IACUC) of Northwestern University.

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
