## [Decision Letter]

Thank you for submitting your article "Upregulation of neurovascular communication through filamin abrogation promotes extraneous cerebral neurogenesis" for consideration by *eLife*. Your article has been reviewed by two peer reviewers, and the evaluation has been overseen by a Reviewing Editor, Marianne Bronner, and Sean Morrison as the Senior Editor. The following individuals involved in review of your submission have agreed to reveal their identity: Alex Kuan (Reviewer #2).

The reviewers have discussed the reviews with one another and the Reviewing Editor has drafted this decision to help you prepare a revised submission.

Summary:

This manuscript describes how filamin abrogation induces a phenotype reminiscent of that found in a human disorder present in females that lose one copy of the X-linked Filamin A gene. The phenotype includes periventricular ectopic neuronal nodules. As in the human condition, the development and structure of the cortex is normal, indicating that the presence of ectopic neurons in those nodules is not the result of defective migration.

This is a nicely done study that provides a good model for the human disorder. It would be useful if the authors could provide a deeper understanding of the endogenous role of filamin or the pathological mechanism that initiates the defects. Open questions that would be worth addressing include: (i) insights into the mechanism that initiates this ectopic growth, (ii) explanations of why the phenotype develops so late in development and importantly, (iii) given that the loss of filamin is achieved by the use a filamin floxed alleles using Emx and Nes Cre driver lines, why the ectopic nodules are sporadic.

Essential revisions:

1) The expression pattern of filamin A and filamin B in mouse embryos, at E12 and E14-15 must be shown.

2) Figure 7 showed convincing induction of VEGFa in the mutant embryonic cortex by immunoblot, but it did not provide spatial information. The author may consider using in-situ hybridization against VEGFa to test whether increased mRNAs coincide with the periventricular nodules.

3) Despite homogenous gene deletion, why is periventricular neurogenesis and ectopia a sporadic event rather than continuous? Secondly, why do these extra neurons apparently not engage in upward radial migration to disrupt the cortical lamination? Is that because of the disruption of local radial glial scaffold? Is that part of the ectopic EMT consequence? These issues should be discussed.

4) Rather than using the term "extraneous" (irrelevant?) cerebral neurogenesis in the title and throughout the manuscript, the authors should consider using a less colorful term, such as "ectopic periventricular" neurogenesis?

5) The statement that Tbr2-GFP analysis demonstrates that PH neurons are daughters of Tbr2+ IPs is not correct. Tbr2 is also transiently expressed in neurons born from RGCs. Furthermore, the BrdU analysis presented is not demonstrative that ectopic neurons are derived from IPs. Cell cycle exit analysis does not distinguish IPs from RGCs. In fact, ectopic neurons may come from RGCs, which are also ectopic. Expression analysis shows that the primary effect of filamin loss is an increase in EMT behavior of RGCs (E13/E14) rather than in IPs.

6) IPs exist in mouse cortex since E11. Therefore, if the PH phenotype is due to defects in IPs why it is only evident at E14 and only affects upper layer neurons? If the loss of filamin is initiated at later stages, would this lead to the same phenotype? This can be tested using another Cre line or by an electroporation of Cre construct.

7) IPs and Ns appear in VZ simultaneously, and alterations also involve VZ, such that the entire VZ + SVZ bend coherently toward ventricle (Figure 3). Which is the primary defect? What triggers the onset? Are defects in vasculogenesis also simultaneous, or they appear after ectopia is initiated?

8) The description of the phenotype as an activation of EMT is unnecessary. The affected cells are not epithelial nor are they converted into mesenchymal cells. It is clear that Snail (and Twist) are activated and repress components of cell adhesion complexes, but this does not mean that this is an EMT process.

9) Along the same lines, the authors mention that "loss of epithelial characteristics of RGCs represents a transition in their identity". Even considering that this is the case, it is not clear to which identity the cells would transition to. Is AJ loss secondary to IP + neuron intrusions into the ventricle?

10) Given that the loss of filamin is a pathological condition, what is the endogenous role of filamin? What is the mechanism of Snail/Twist activation upon filamin loss? Is filamin a Snail repressor?

11) Why IPs go to VZ, instead of moving further basal? Is there any attractive cue or a particularly favorable microenvironment for them to go/stay at the VZ?

12) Why the rosette phenotype leads to PH rather than subcortical band heterotopia (SBH) as in other mutant models? What is the mechanism of rosette formation? The authors suggest that this may be through a reduced stiffness of the VZ but it is not clear how this has been measured. What type of cell behavior would give rise to these rosettes?

---

## [Author Response]

*Essential revisions:*

*1) The expression pattern of filamin A and filamin B in mouse embryos, at E12 and E14-15 must be shown.*

The expression pattern of Flna and Flnb in mouse embryos at E12.5, E14.5, and E16.5 was analyzed and shown by a previous paper in which I was the co-first author (Sheen and Feng et al., HMG, 2002). We showed that Flna and Flnb were co-expressed in NPCs throughout the course of cortical development and were able to form a heterodimer. Citation of this reference is now included to support the functional redundancy of filamin A and filamin B in cortical NPCs.

*2) Figure 7 showed convincing induction of VEGFa in the mutant embryonic cortex by immunoblot, but it did not provide spatial information. The author may consider using in-situ hybridization against VEGFa to test whether increased mRNAs coincide with the periventricular nodules.*

We followed this excellent suggestion, combined in situ hybridization with IHC analyses, and now present evidence for the specific association of increased Vegfa mRNA with loss of RG adhesion and periventricular angiogenesis (which are the two pivotal events that lead to PH) in the revised Figure 7. Vegfa mRNA signals faded a little during the subsequent immunostain, but the information should be very clear.

*3) Despite homogenous gene deletion, why is periventricular neurogenesis and ectopia a sporadic event rather than continuous? Secondly, why do these extra neurons apparently not engage in upward radial migration to disrupt the cortical lamination? Is that because of the disruption of local radial glial scaffold? Is that part of the ectopic EMT consequence? These issues should be discussed.*

The sporadic nature of periventricular neurogenesis is due to the essential vascular contribution and determined by the location of abnormal blood vessels that are not distributed continuously. As neurons in PH were generated as a surplus, they did not belong to the neocortex and might not have the correct cues or scaffold for upward migration. Additionally, RG defects only affected the apical VZ and did not appear to disrupt the bulk of basal RG fibers to disturb the normal neuronal migration as demonstrated in the newly added Figure 4—figure supplement 1. While the EMT-like changes lead to substantial alteration in cell’s structure and biochemical context, it was a confined local event in the apical VZ, which may not affect those IPs that were already fate-restricted to generate neocortical neurons before E13.5. We appreciate these questions and added the possible early fate-restriction of late-born cortical neurons to the Discussion.

*4) Rather than using the term "extraneous" (irrelevant?) cerebral neurogenesis in the title and throughout the manuscript, the authors should consider using a less colorful term, such as "ectopic periventricular" neurogenesis?*

We originally used the term “extraneous” and meant to indicate “additional” or “extra” generation of neurons. Thanks for the correction. But we are not sure if “ectopic” is precise either, as it only indicates the location but not the amount. To avoid confusion we chose “surplus” in the revised title. We also revised the text to indicate that there is a net gain of neurons. Please advise if this is still unclear.

*5) The statement that Tbr2-GFP analysis demonstrates that PH neurons are daughters of Tbr2+ IPs is not correct. Tbr2 is also transiently expressed in neurons born from RGCs. Furthermore, the BrdU analysis presented is not demonstrative that ectopic neurons are derived from IPs. Cell cycle exit analysis does not distinguish IPs from RGCs. In fact, ectopic neurons may come from RGCs, which are also ectopic. Expression analysis shows that the primary effect of filamin loss is an increase in EMT behavior of RGCs (E13/E14) rather than in IPs.*

We fully agree that Tbr2 is not expressed exclusively by IPs and it is necessary to clarify how the periventricular neurons were not generated from abnormal RGCs directly. In the revised Figure 3—figure supplement 2, we now include data to show that in the affected (PH^+^) cortical columns, RGs (marked by Pax6) were very scarce and showed little proliferation activity (BrdU and phospho-Histone H3 Ser10). Therefore, the ectopic neurons were unlikely produced by the defective RGs, and the BrdU labeling and cell cycle exit activities in PH^+^ regions were largely from aplicalized IPs. Although the mutant RGs showed changes resembling abnormal EMT, it did not appear to make them divide more. In fact, the contribution of EMT to cancer stem cell and metastasis is currently controversial, and cells that underwent EMT (mutant RGs and MDCK cells following TGFbeta treatment) showed reduced proliferation in our observations.

*6) IPs exist in mouse cortex since E11. Therefore, if the PH phenotype is due to defects in IPs why it is only evident at E14 and only affects upper layer neurons? If the loss of filamin is initiated at later stages, would this lead to the same phenotype? This can be tested using another Cre line or by an electroporation of Cre construct.*

We appreciate this excellent question and performed additional experiments to clarify this issue. IPs exist since E11, but only come into close contact with the blood vessels (which our evidence indicates is needed to induce PH) after E12.5-E13.5 as shown by the newly added Figure 6—figure supplement 1. We hope this new figure answers the question and strengthens our finding of the vascular contribution to PH-genesis. We also added data to show that abrogating Flna in later development by GFAP-Cre (with onset of Cre expression at E13.5) can also induce PH, which supports our conclusion that the phenotype originates from NPCs/RGs but relies on the positive feedback between abnormal RGs and blood vessels.

*7) IPs and Ns appear in VZ simultaneously, and alterations also involve VZ, such that the entire VZ + SVZ bend coherently toward ventricle (Figure 3). Which is the primary defect? What triggers the onset? Are defects in vasculogenesis also simultaneous, or they appear after ectopia is initiated?*

As the conditional knockout occurs in NPCs, the cell autonomous NPC/RG defect should be the primary one, however, it relies on the non-cell autonomous vascular trigger and amplification, which forms a positive feedback loop as illustrated in the model of Figure 7: The loop is initiated by the failure of down-regulating the vascular signals from the mutant RGs, then results in a local change in IPs’ localization and microenvironment. We made text revisions to state this positive feedback more clearly.

*8) The description of the phenotype as an activation of EMT is unnecessary. The affected cells are not epithelial nor are they converted into mesenchymal cells. It is clear that Snail (and Twist) are activated and repress components of cell adhesion complexes, but this does not mean that this is an EMT process.*

We have revised “EMT” to “EMT-like” for the following reasons: 1) derived from neuroepithelial, RGs retain many key features of epithelial cells such as polarity and adhesion; 2) what occurred to the mutant RGs were not only loss of adhesion and Snail/Twist expression but also increased mesenchymal ECM (HA and FN), which fulfills the definition of EMT; 3) the MDCK model demonstrates that functional deficiency of *FLNA* increases EMT. We hope this revision is acceptable.

We feel the EMT-like phenotype is essential for PH as it could cause a fundamental change in the local ECM (besides FN and HA) and contribute significantly to the microenvironment alteration in addition to vascular cues. Please also see the response to point 9.

*9) Along the same lines, the authors mention that "loss of epithelial characteristics of RGCs represents a transition in their identity". Even considering that this is the case, it is not clear to which identity the cells would transition to. Is AJ loss secondary to IP + neuron intrusions into the ventricle?*

The neurogenesis of IPs should be secondary to RG defects, which we think resemble EMT-like changes. These RG defects include loss of normal adhesion, polarity, and Pax6 expression but increase in the expression of Snail, Twist, FN, and HA. These mutant RGs were clearly different from normal RGs, which made us believe they went through a transition in identity. While it would be interesting to further delineate the full cell molecular characters of these defective RGs in future studies, it is clear that they could no longer maintain the VZ and SVZ compartmentation, which resulted in the abnormal localization, proliferation, and neurogenesis of IPs near the ventricle.

*10) Given that the loss of filamin is a pathological condition, what is the endogenous role of filamin? What is the mechanism of Snail/Twist activation upon filamin loss? Is filamin a Snail repressor?*

We appreciate these questions and stated more clearly that the physiological function of filamin/*FLNA* is acting as a molecular break to prevent excessive signals between NPCs and the developing vasculature in order to maintain homeostasis at tissue level. Since there is so far no evidence on the direct role of filamin in transcription regulation, we believe elevated Snail/Twist reflected abnormal signal transduction and sustained EMT-like events of the mutant RGs.

*11) Why IPs go to VZ, instead of moving further basal? Is there any attractive cue or a particularly favorable microenvironment for them to go/stay at the VZ?*

We believe the mislocalization of IPs from SVZ to VZ was secondary to RG defects. Mild adhesion defect of RGs were also observed (in a low frequency) in some Flna_cKO_Flnb_WT_ embryos where IP mislocalization was not obvious. However, the variable localization of IPs and sporadic nature of the phenotype made it difficult to analyze this mutant quantitatively. In contrast, the mislocalization of IP was found to coincide with RG defects exclusively. This suggests that RGs are essential for the proper SVZ localization of IPs. Since the RG defect in the filamin mutant is not purely “structural,” it is possible that the IPs were also trapped to the periventricular space biochemically by altered ECM and vascular cues as discussed in response to points 8 and 9.

*12) Why the rosette phenotype leads to PH rather than subcortical band heterotopia (SBH) as in other mutant models? What is the mechanism of rosette formation? The authors suggest that this may be through a reduced stiffness of the VZ but it is not clear how this has been measured. What type of cell behavior would give rise to these rosettes?*

Our data suggest that the rosette-like structure in filamin mutants was due to EMT-like defects instead of a simple loss of cell adhesion, morphology or scaffold function of the mutant RGs. These RGs were no longer Pax6+ but instead provided altered ECM, changed vascular cues, and promoted extra local IP proliferation and neuron production, which may together physically push the adjacent unaffected RGs and lead to their invagination or rosette formation. Therefore, the formation of rosette is the by-product but not the prerequisite for PH, it did not interfere with neocortical neuronal migration and thus was different from the neuronal migration arrest that results in SBH. In theory, the formation of epithelial rosette is largely driven by increased acto- myosin-dependent apical constriction and changes in epithelial morphology in normal morphogenesis. But here we face a situation in which the rosettes formed around relatively normal RGs due to defects in their neighboring cells that lost apical constriction. We agree that it was somewhat speculative to talk about the tissue stiffness since we did not measure it. To address this extremely helpful critique, we made text revision to present more clear and objective conclusions.